# Plant Extracts for Production of Functionalized Selenium Nanoparticles

**DOI:** 10.3390/ma17153748

**Published:** 2024-07-29

**Authors:** Krystyna Pyrzynska

**Affiliations:** Department of Chemistry, University of Warsaw, Pasteur Str. 1, 02-093 Warsaw, Poland; kryspyrz@chem.uw.edu.pl

**Keywords:** selenium nanoparticles, green synthesis, plant extracts, applications

## Abstract

In recent years, selenium nanoparticles (SeNPs) have attracted expanding consideration, particularly in the nanotechnology field. This element participates in important biological processes, such as antioxidant defense, immune function, and thyroid hormone regulation, protecting cells from oxidative damage. Selenium in the form of nanoscale particles has drawn attention for its biocompatibility, bioavailability, and low toxicity; thus, it has found several biomedical applications in diagnosis, treatment, and monitoring. Green methods for SeNP synthesis using plant extracts are considered to be single-step, inexpensive, and eco-friendly processes. Besides acting as natural reductants, compounds from plant extracts can also serve as natural capping agents, stabilizing the size of nanoparticles and contributing to the enhanced biological properties of SeNPs. This brief overview presents the recent developments in this area, focusing on the synthesis conditions and the characteristics of the obtained SeNPs.

## 1. Introduction

The fascinating properties of nanostructured materials, such as their size, large surface area to charge ratio, reactive surface, bioactivity, possibility of functionalization, and optical and electronic properties, have made them favorable in many fields of industry and medicine [1,2]. Among these applications, nanoparticles are very useful in biomedical applications to treat, diagnose, monitor, and regulate biological and targeted drug delivery systems in current therapies [3,4]. In recent years, selenium nanoparticles (SeNPs) have attracted expanding consideration. This element plays a key role in several major metabolic pathways, such as protecting cells from oxidative damage and infection, thyroid hormone metabolism, and immune function [5,6]. Selenium deficiency has been linked to a range of serious diseases, like various forms of cancer, diabetes, arthritis, and muscular dystrophy [7,8,9].

The properties of SeNPs depend on their synthesis method, the kind of reagent used for selenium salt reduction, the use of some additives, reaction temperature, and time [10,11,12,13,14]. The synthesis steps involve physical, chemical, and biological methods. Physical methods for SeNP synthesis (hydrothermal, laser ablation, vapor deposition, microwave irradiation, ultrasonication, and electrodeposition) have been proposed, but high cost and high energy consumption are their main disadvantages for wide application [11,14,15]. The chemical approach for SeNP synthesis is based on the reduction of inorganic selenium forms by different reducing agents (ascorbic acid, glucose, cysteine, and glutathione), usually in the presence of a stabilizing agent to prevent the aggregation of nanoparticles. However, some residuals of the used chemicals limit the application of the formed SeNPs in pharmaceutical and medicinal areas. Biological methods of SeNP synthesis involve the use of plant extracts or microorganisms (bacteria, fungi, and algae). According to the reviewed literature, nanoparticles produced by such green synthesis demonstrate greater compatibility with human organs and tissues. The mechanism of SeNP biosynthesis using microorganisms varies among diverse microbial species and includes several metabolic pathways with different enzymes for the reduction process. It also requires time-consuming procedures for maintaining and keeping cell cultures. Previous reports have highlighted the advantages of green synthesis over chemical and physical methods [15,16,17,18,19]. The use of plant extracts in the synthesis of SeNPs does not require extreme conditions. Moreover, it is economical and eco-friendly because the reducing agents are easily available and many of these plants possess traditional and medical uses [14]. The typical synthesis combines the plant extract with selenium precursor at different ratios [20]. Plant extracts contain phenolic acids, flavonoids, terpenes, coumarins, stilbenes, tannins, polysaccharides, proteins, and other reducing substances. Most of these molecules contain hydroxyl or phenol hydroxyl groups. Their oxidation reactions can be expressed as follows:R(OH)*_n_* = *n*H^+^ + *n*R = O + *n*e

The reduction of selenite takes place as:SeO_3_^2−^ + 6H^+^ = Se^0^ + 3H_2_O

During the reaction, the color changes from colorless to red, indicating SeNP formation. Then, a solution is stirred at room or elevated temperatures for various periods, centrifuged, and the nanoparticles are washed, dried, and collected. It should be noted that several studies have proposed the synthesis of SeNPs conducted for long reaction times. For example, *Brassica oleracea* extract was incubated with Se salt for 2–3 days [21], and *Abelmoschus esculentus* extract was incubated for 2 days [22]. Conversely, the reaction with mandarin peel-derived pectins combined with olive pomace extract was completed in 20 min, indicated by a fast decrease in selenium content in the solution [23].

Besides acting as natural reductants, plant extract compounds can serve as natural capping agents, preventing clusters or aggregates of nanoparticles and stabilizing their size. The final morphology of NPs largely depends on the capping agent that is adsorbed or bound on their surface [24]. This additional layer on nanoparticles is also an essential tool in nanomedicine and contributes to the enhanced biological properties of SeNPs, such as biocompatibility and bioavailability [25,26]. The phenolic compounds present in plant extracts, besides their high antioxidant activity, are electron donors due to breaking O-H bonds in their enol groups. Thus, they can reduce selenium salts to elemental form with their related oxidation to the corresponding quinones [13,27]. Terpenes, a large group of compounds used in folk medicine, are often involved in the odors released by plants. They hold antioxidant, anti-inflammatory, antiseptic, antiplasmodial, and many other properties [28]. Historically utilized in the textile industry, tannins exhibit antioxidant, antibacterial, antiviral, and antiparasitic properties [29]. Proteins with low concentrations in plant extracts cannot participate in nanoparticle synthesis but have great importance for their stabilization and prevention of aggregation due to the affinity of binding metals with the carbonyl group of amino residues [30].

Different plant material extracts have been recently proposed for the biosynthesis of SeNPs, such as herbs and other medicinal plants [27,31,32,33,34,35,36,37], teas with different processing methods [38,39,40,41], coffee beams [42,43], plants found only locally [44,45,46,47], some fruits and vegetables [48,49,50,51,52,53,54], and also their agro-industrial waste [55,56,57,58,59]. Biomolecules and secondary metabolites from different parts of these plants can significantly reduce Se salts to Se elementary form. Easy accessibility as well as traditional and pharmacological uses are probably the determining factors for the choice of a given plant in a synthesis process. There is a maximum probability of succeeding by using plant materials with high amounts of polyphenolic compounds. For this reason, the contents of these compounds are often determined by chromatographic analysis. The Folin–Ciocalteu assay is also performed to determine the so-called total content of polyphenolic compounds, although it measures a given sample’s reducing properties [60]. However, the presence of other compounds in plant extracts could affect (positively or negatively) the efficiency of nanoparticle synthesis. Ascorbic acid is an example of such a compound due to its reductive properties [61]. In some proposed procedures, this compound is added to help start the reduction reaction. Moreover, the chemical composition of polyphenolic compounds in plants may vary across growing seasons, climates, and horticultural practices, thus affecting the properties of the synthesized nanoparticles [21]. The use of plants that are available only locally limits large-scale global production. For example, SeNPs were synthesized using extracts from *Angelica sinensis*, which grows in cool high-altitude mountains in East Asia [46] or *Pluchea indica* from the tropical climate of Southeast Asia and the Pacific Islands [44].

Research interest concerning the synthesis and applications of SeNPs has increased in recent years and several review articles have focused on this topic. These papers mostly discuss various synthesis methods employed for selenium nanoparticle preparation and their advantages and disadvantages over each other [15,62,63] or examine only the conditions in biological approaches [12,13,18]. Most works are dedicated to the pharmacological potential of SeNPs in biomedical research and toxicology studies [11,13,14,19,24,64]. Some articles highlight the key role of selenium nanoparticles in treating particular diseases, like cancer [65,66], diabetes [8,67], infection [68], and brain diseases [69]. The factors affecting the synthesis of selenium nanoparticles using plant extracts, which greatly affect their use, are seldom discussed [16]. Thus, this brief overview focuses on the recent developments in SeNP plant-mediated synthesis and compares the obtained nanoparticle parameters.

## 2. Characterization of SeNPs

The important parameters for synthesized nanoparticles are their size and distribution, shape, surface charge and area, degree of aggregation, zeta potential, and a variety of other common and more specific properties [70]. SeNPs need to possess particular structural and functional properties for use in diverse applications. For example, a characterization strategy for biomedical applications also includes sterility and pyrogenicity assessment, biodistribution (absorption, distribution, metabolism, and excretion (ADME)), and toxicity [71]. Thus, synthesized SeNPs have been characterized using several microscopic and spectroscopic methods [72,73].

### 2.1. UV–Vis Spectra of SeNPs

SeNPs exhibit characteristic UV–Vis absorption spectra in the range of 200–500 nm, and the surface plasmon vibration of the excitation state of SeNPs is responsible for the appearance of the dark red color in the solution [74]. Different studies have reported various locations for the absorption peak: SeNPs synthesized from the extract of *Dispropyros montana* bark at 255 nm [59], *Abelmoschus esculentus* fresh plant at 298 nm [23], peel *Solanum melonga* L. extract at 320 nm [58], walnut leaf extract at 375 nm [33], green tea at 400 nm [38,41], *Moringa oleifera* extract at 445 nm [37], and ginger extract at 550 nm [61]. In some cases, the intensity of the absorption peak increases with an increase in reaction time. It also shifts to a longer wavelength, pointing to an increase in particle size due to aggregation, as shown in Figure 1A for SeNPs synthesized using *Moringa oleifera* extract [75].

Besides prolonged time of synthesis, thermal treatment can cause changes in the position of the characteristic absorption peak and its intensity, as was reported by Tarmizi et al. [37]. The absorbance wavelength increased with rising temperature from 446 nm at 4 °C, to 453 nm at 25 °C, up to 474 nm at 50 °C (Figure 1B). Thus, heating of the reaction mixture increased the size of the SeNPs. However, too high a temperature could affect the properties of SeNPs due to the degradation of phytochemicals involved in the synthesis reaction.

### 2.2. Particle Size, Distribution, and Morphology

The location of an absorption peak can be used for a rough estimation of nanoparticle size [75]. It was proved that a suspension of SeNPs of 20 nm diameter exhibited an absorption maximum below 250 nm. With an increase in the size of the nanoparticles, the absorption maximum shifted toward higher wavelengths. For SeNPs with the size of 100 nm, it was located at around 350 nm, and with the size of 240 nm, it was observed at 680 nm. Scanning emission microscopy (SEM), transmission electron microscopy (TEM), and dynamic light scattering (DLS) measurements are often used for the examination of SeNP size, as well as their shape, agglomeration, and distribution. Disagreement between the sizes obtained by these methods is often observed because TEM analyzes the metallic core, while DLS measures the hydrodynamic volume of the particles, which includes the hydration layer and other possible stabilizers, leading to a larger overall size.

Selenium nanoparticles synthesized by plant extracts are represented by spheres or nanorods. Spherical SeNPs were observed by TEM with a particle size diameter of 1–3 nm using *Theobroma cacao* L. bean shell extract as a stabilizing and capping agent [55], which was at least one order of magnitude lower than those of other reported SeNPs. On the other hand, DLS analysis of SeNP particle size obtained from *Citrus limon* fruit showed a range of 1100–3500 nm [50]. The size of SeNPs synthesized with *Moringa oleifera* leaves also differed greatly, wherein the particle size ranged from 20 to 250 nm, as revealed by SEM analysis [37]. The observed differences could be due to the presence of different compounds in each of the extracts used in the synthesis process as well as their content. Moreover, highly aggregated solutions have much larger hydrodynamic diameters measured by the DLS method than those obtained by TEM analysis, as mentioned above. However, most of the recent reports describe selenium nanoparticles not exceeding 100 nm. It should be recognized that it is more difficult to control the course of green SeNP synthesis than chemical synthesis. Plant-derived compounds could affect the mechanism of this reaction in different ways and thus the properties of nanoparticles [31,42,45].

A study concerning the application of leaf extracts from several medicinal plants (blackberry, hop, lemon balm, raspberry, and sage) conducted under the same synthesis conditions showed that the resulting products varied in size and shape [27]. Their SEM and TEM images are presented in Figure 2. Another example is the synthesis of SeNPs with the extracts of different kinds of teas (black, green, red, and white) [38]. SeNPs synthesized using the extract of white tea had the smallest dimension (3.94 nm), followed by black tea (4.89 nm), red tea (7.78 nm), and green tea (12.15 nm). The leaves of all studied tea types, derived from the same *Camelia sinensis* plant and from the same producer, were subjected to different processing methods, which could have caused changes in their profiles of polyphenolic compounds acting as selenium reductants.

The size and distribution of SeNPs are often the key to their specific, desired properties and finally to their respective applications [76]. Smaller NPs have a higher surface area to volume ratio as compared to larger ones. This enables higher biological activity, such as bioavailability and biodistribution, allowing them to interact closely with the microorganism cell membrane, facilitating interaction and intracellular diffusion in in vivo applications [77]. According to Golub et al., the target size of SeNPs for dietary supplement applications is 100–200 nm [24]. SeNPs with smaller particle sizes also exhibit higher antioxidant potential through scavenging free radicals or reducing activity [78,79,80].

Cheng et al. investigated the relationship between the size of SeNPs and their availability in soil for agriculture production and found that the mobility of SeNPs is decreased with increasing particle size [81]. Small-size nanoparticles (30 nm) are more bioavailable in plants and soil, have better stability and migration in soil, and reduce aggregation due to greater adsorption of organic matter. The accumulation of these small SeNPs in *Brassica chinensis* leaves promoted photosynthesis and increased the yield of plants. For the production of Se-rich vegetables, SeNP size in the range of 11–631 nm was recommended based on linear regression analysis [81].

The polydispersity index (PDI) is also an important parameter for the characterization of SeNPs. The PDI value estimates the average uniformity of a particle solution [82]. A lower PDI indicates a more homogeneous sample with particles that are more uniform in size. According to Nobbmann, for monodisperse samples, the PDI value is less than 0.1, for moderately polydisperse samples, it ranges from 0.1 to 0.4, and the distribution of nanoparticles is broad with a PDI below 0.4 [83]. However, for the specific requirements of nanoparticles, an appropriate PDI value may be chosen. For example, the PDI values of synthesized SeNPs using popular plant extracts (at Se/extract ratio = 1:1) were increased in the order of green tea (0.253) < black tea (0.317) < chamomile (0.481) < mint (0.944), and the opposite order was obtained when evaluating the ability of these extracts to neutralize hydroxyl radicals [38]. Similar results were recorded in the reaction with the extracts of different kinds of teas [41]. The PDI values amounted to 0.165 for green tea, 0.298 for black, 0.481 for red tea, and 0.381 for white tea. An increase in the plant extract content used in the synthesis and additional heating of the post-processing mixture decreased the homogeneity of SeNPs [27,38,41].

### 2.3. Surface Composition and Charge

Energy dispersion X-ray spectroscopy (EDXS) is performed for the examination of the elemental composition of nanoparticles. The highest content of selenium (82%) in SeNPs was determined when the extract of *Cassia auriculata* was used [47], while SeNPs decorated by *Asteriscus graveolens* components had only 13.9% [45]. Besides selenium, the analysis also revealed the presence of carbon, followed by oxygen and nitrogen, which could correspond to the compounds of the extracts. The process of reduction and decoration of the SeNP surface by the components of the extracts is very often revealed by Fourier-transform infrared (FTIR) spectroscopy [72]. It allows the characterization of the capping ligands and monitoring of the nanoparticle surface composition. The presence of several functional groups, such as -OH, -NH_2_, -C=O, -CO, and -COOH involved in SeNP formation and stabilization, is mainly evaluated. Lazcano-Ramirez et al. [35] found specific bands corresponding to the presence of conjugated C=C, C≡C, and C=O ester fatty acid groups, besides the common hydroxyl, carbonyl, and carboxylic groups. The presence of amide groups was confirmed in the synthesis of SeNPs using *Asteriscus gravelenes* components [45], citrus fruit [48], *Vaccinium arctostaphylos* fruit [49], and onion extract [53].

The surface properties of SeNPs can greatly impact their interaction with biological pathways [84,85,86,87,88]. Compounds from plant extracts cover selenium nanoparticles with a layer consisting of polyphenols, proteins, lipids, polysaccharides, amino acids, or sugars. In addition, these phytocompounds can control the growth of particles in three dimensions, giving them a spherical shape [89]. Memon et al. [90] suggested that polysaccharides are better for SeNP synthesis than polyphenols or proteins as enzymes can degrade these compounds, resulting in the aggregation of nanoparticles in the acidic environment of a stomach. However, the main role of capping agents is to create the appropriate and desired biological function of SeNPs. Biological moieties enhance cytotoxicity against various cancer cell lines, increase stability, and help to target drug delivery without non-specific interaction compared to nanoparticles alone [91,92,93,94]. Functionalization of SeNPs can significantly improve their direct antiradical and reducing properties [24,94].

Experimentally, surface charge is characterized by measuring electrophoretic light scattering, and the results are presented as the zeta potential (ZP) [73]. Its value describes the nanoparticles’ surface charge and determines their long-term stability. Generally, suspensions of nanoparticles with values for zeta potential from −20 mV to +30 mV or higher are to be considered stable due to electrostatic repulsion between strongly charged surfaces. In the case of lower-charged nanoparticles, their aggregation occurs due to van der Waals attraction forces [63]. Most of the literature focused on the synthesis of SeNPs using plant extracts reports values of zeta potential that ensure stability (Table 1). The highest zeta potential value, distinguishing nanoparticles obtained with okra flowering plant extract, is equal to −64 mV [23]. For SeNPs stabilized by chitosan derivatives, a high positive charge (zeta potential of 46.73 mV) ensures the stability of nanoparticles in an aqueous medium and the possibility of additional ionic interaction with a negatively charged cell membrane [95].

The value of the zeta potential can be affected by the properties of nanoparticles but also by pH and ionic strength. For example, at low pH values, the ionization of some functional groups, e.g., carboxylic groups, can decrease, resulting in a lower zeta potential [80,96,97]. For SeNP synthesis using polysaccharides, polyphenols, or peptides, the optimal reduction of selenium salts mostly occurs at a pH of about 4. However, photofabrication of SeNPs synthesized with *Moringa oleifera* leaves was conducted at pH 8–9, indicating that the functional groups in the extract compounds were more active in a weakly alkaline environment [37].

Golub et al. evaluated the properties of SeNPs stabilized with raw and purified mandarin peel-derived pectins and added olive pomace extract for functionalization [24]. Raw polysaccharides have been found to exhibit higher stabilizing effects on SeNPs compared to purified ones. Their average diameter during 30 days of storage decreased in the order: SeNPs coated with raw pectins (84.7 nm decrease in size) > raw pectins + olive pomace extract (38.6 nm) > purified pectins + olive pomace extract (38.1 nm) > purified pectins (36.0 nm) (Figure 3). However, the zeta potential values (ranging from −22.3 to −23.1 nm) were not significantly affected by the type of pectin or additional functionalization.

Table 1 presents the recent literature reports on synthesizing SeNPs using plant extracts with their characteristics and properties.

**Table 1 materials-17-03748-t001:** The recent literature data on SeNP synthesis using plant extracts.

Plant Material	Synthesis Conditions	Characteristics	Properties	Ref.
Broccoli (*Brassica oleraccea*) leaves	30 mM Na_2_SeO_3_ + extract (30 mL), stirred for 2–3 days	10–28 nm (SEM) average 15.2 nm (TEM)	Antioxidant, anticarcinogenic	[21]
Okra (*Abelmoschus esculentus*)	Na_2_SeO_3_ (0.08 g dissolved in 50 mL of water) + extract stirred for 48 h	17.3 nm (DLS) 46.15 nm (TEM) ZP: −64 mV	Antibacterial	[22]
Mandarin peel-derived pectins functionalized with olive pomace	0.1 M Na_2_SeO_3_ (1 mL) + 5 mL of 1% olive pomace (5%) + pectins (15 mg), stirred for 20 min	171–217 nm PDI: 22.7 ZP: −22.5 mV	Antioxidant	[23]
Herbs (lemon balm, hop, raspberry, sage, blackberry)	0.1 M Na_2_SeO_3_ (2.5 mL) + extract (2.5 mL), stirred for 60 min	74.0–96.8 nm PDI: 0.103–0.132	Antibacterial, antioxidant	[27]
Walnut leaves	0.01 M Na_2_SeO_3_ (15 mL) + extract (5 mL), heated with microwaves (800 W) for 4 min	208 nm PDI: 0.206 ZP: −24.7 mV	Antibacterial	[33]
*Withania somnifera*	0.050 M Na_2_SeO_3_ + extract (100 mL)	45–90 nm	Antioxidant, photocatalytic	[34]
*Amphipterygium glaucum* leaves	0.01 M Na_2_SeO_3_ (10 mL) + extract (80 µL), stirred for 24 h at 40 °C	8.0 nm PDI: 0.236	Antifungal	[35]
*Crocus caspius*	Na_2_SeO_3_ (17.3 g in 100 mL) + extract (5 mL), stirred for 48 h	average 23.47 nm ZP: −44.75 mV	Antimicrobial, antifungal, photocatalytic	[36]
*Moringa oleifera* leaves	0.05 M Na_2_SeO_3_ (5 mL) + extract (20 mL), stirred for 48 h at 37 °C	20–250 nm	Antioxidant, antidiabetic	[37]
Black and green tea, herbs (chamomile, mint)	0.1 M Na_2_SeO_3_ (2.5 mL) + extract (2.5 mL), stirred for 60 min	54.8–108 nm	Antioxidant	[38]
*Lycium barbarum* + green tea	25 mM Na_2_SeO_3_ (0.5 mL) + extract (2 mg/L) + 1 mL of tea infusion, dialyzed overnight	average 260 nm PDI: 0.242ZP: −24.1 mV	Antioxidant, neuroprotective agent	[40]
Black, green, red, and white tea	0.1 M Na_2_SeO_3_ (2.5 mL) + extract (7.7 mL), stirred for 60 min	3.9–12.5 nm PDI: 0.165–0.381	Antioxidant	[41]
*Elaeagnus indica*	50 mM of H_2_SeO_3_ + extract (200 mL), stirred for 24 h	av. 14 nm	Antimicrobial, photocatalytic	[42]
*Asteriscus graveolens* aerial parts	0.01 M H_2_SeO_3_ (25 mL) extract (75 mL), incubated for 24 h	21.6 nm PDI: 1.00 ZP: −24.1 mV	Anticancer	[44]
*Vaccium artostaphylos* L. fruits	0.1 M Na_2_SeO_3_ (9 mL) + extract (1 mL), stirred for 24 h	average 50 nm (SEM) 246 nm (DLS) PDI: 0.267 ZP: −11.5 mV	Antibacterial	[49]
Lemon and grapefruit juice and peels	Na_2_SeO_3_ (8–12 mM) + extracts, pH 7, stirred at 70 °C for 2 h	1100–3500 nm (DLS) PDI: 0.127	Antibacterial	[50]
Ginger and onion	Na_2_SeO_3_ (10 g) + extract (100 mL), stirred at 60 °C for 3–12 h	90–114 nm	Antimicrobial	[52]
Cacao bean shell (*Theobroma cacao* L.)	Na_2_SeO_3_ (0.14 g) + extract (50 mL), heated in the microwave oven (788.6 W) for 15.6 min	1–3 nm	Antioxidant	[54]
*Diospyros montana* bark	0.3 M Na_2_SeO_3_ + 10 mL of extract, stirred for 24 h	120–200 nm (SEM) 20–200 nm (TEM) 140.4 nm (DLS) PDI: 0.418	Antioxidant, antibacterial, antiproliferative	[59]
*Terminalia arjuna* bark	0.35 M of Na_2_SeO_3_ (10 mL) + extract (10 mL), stirred for 24 h at 37 °C	100–150 nm ZP: −26.1 mV	Antioxidant, antimicrobial, anticancer	[94]
*Orthosiphon stamineus* leaves + curcumin	20 mM of Na_2_SeO_3_ (45 mL) + 5 mL of extract + curcumin (5 mg/mL), stirred for 30 min	100 nm	Tissue engineering	[95]
*Hibiscus esculentus* L.	0.01 M Na_2_SeO_3_ + extract (10 mL), stirred for 24 h at 45–50 °C	50.1 nm (SEM) 266.3 nm (DLS) ZP: 51.3 nm	Anticancer, antibacterial, antifungal	[98]

SEM: scanning emission microscopy; TEM: transmission electron microscopy; DLS: dynamic light scattering; PDI: polydispersity index; ZP: zeta potential.

## 3. Applications

The potent actions of SeNPs from plant-mediated synthesis advise their successful employment in various disciplines. They have exceptional physicochemical properties, such as low toxicity, biocompatibility, and chemical stability [99]. Moreover, the preparation of SeNPs using medicinal plants may enhance their beneficial properties.

The biological and pharmacological properties of selenium nanoparticles have been extensively studied to reveal their antioxidant, anti-inflammatory, anticancer, antimicrobial, antidiabetic effects, protective effects against cardiovascular disorders, and neurodegenerative properties, among others. Particularly, their ability to cross cell membranes for drug administration, biocompatibility, and low toxicity make them useful in a variety of biomedical applications in diagnostics and therapy [11,19,67]. The antioxidant and anti-inflammatory activities of selenium nanoparticles can be exploited to prevent or reduce damage to different tissues caused by ionizing radiation exposure [100,101,102]. The radioprotective effects of SeNPs in irradiation-induced nephropathy were higher than those of sodium selenite [100]. The physical and chemical properties of SeNPs provide opportunities to develop electrochemical sensors to detect biologically relevant analytes such as hydrogen peroxide, heavy metals, or glucose [103]. They have also generated considerable interest in food science for applications in nutritional supplements, as food additives, and fabrication of active food packaging preserving the safety and quality of food products [104,105,106].

This section describes the main activities of SeNPs produced by plant-mediated synthesis and highlights their recent applications in the biomedical field, such as cancer, diabetes, and microbial infections. Interested readers can find more specific information focused on the biological activities of SeNPs and their applications in recently published reviews [13,14,15].

### 3.1. Antioxidant Activity

Several researchers have examined the antioxidant activities of SeNPs using different in vitro chemical assays or in vivo cell-based methods. SeNPs play a role in the direct neutralization of reactive oxygen species, regulating their content produced during biochemical reactions, and protecting cells from oxidative stress and damage. They can also improve the activity of antioxidant enzymes, such as superoxide dismutase, catalase, or glutathione peroxidase, with equal efficiency in comparison to other selenium species, but with less toxicity [12]. Increased oxidative stress is associated with the development of several human diseases, like cancer, diabetes, and cardiovascular and neurological diseases. Some studies reported higher antioxidant activity of plant-synthesized SeNPs than the plant extract used for this reaction [24,32,35,54,68]. However, it is important to remember that even though selenium is an essential trace element for humans, its beneficial and harmful effects depend on the dose [5]. At low concentrations, it serves as an antioxidant, while at high concentrations, it causes toxicity, serving as a pro-oxidant. The pro-oxidant activity of selenium is mainly incorporated in the treatment of cancer due to differential activity between cancer and normal cell lines.

SeNPs synthesized with *Moringa oleifera* leaves and *Disiospropos montana* bark extracts were more powerful in their antioxidant activity to scavenge DPPH radicals than to reduce ferric ions [37,59]. SeNPs from green tea extract capped with *Lycium barbarun* polysaccharides demonstrated dose-dependent antioxidant activity to neutralize DPPH radicals and a neuroprotective role against H_2_O_2_-induced oxidative stress [40]. In a study, Abd-Elaraoof et al. reported that the nanocomposite of SeNPs@*Posidonia oceanica* extract@chitosan possessed higher antioxidant activity than standard ascorbic acid [107].

Every form of selenium has anticancer activity, more or less, but these activities depend also on the dose as well as the type of cancer and stage of disease [108]. Several anticancer effects of SeNPs have been suggested, such as the formation of reactive oxygen species, apoptosis, cell cycle arrest, modulation of intracellular redox status, and interruption of the cell signaling pathway [19,67,91]. The anticancer mechanism of SeNPs is still under investigation, but it was proved that they selectively accumulated inside malignant cells, inhibiting their growth with minimal side effects on normal cells. Higher antitumor bioactivity and lower cytotoxicity were determined for SeNPs compared to other selenium species [109].

Zeebaree et al. [45] reported that SeNPs decorated by *Asteriscus graveolleus* components exhibited high anticancer activity by inducing apoptosis of human liver carcinoma HepG2 cells, with an IC_50_ value of 3.98 µg/mL, while IC_50_ for this plant extract was 5.61 µg/mL. The MTT assay revealed high growth control against human gastric cancer AGS cell lines and human breast adenocarcinoma MCF-7 cell lines, with IC_50_ values of 50.7 and 47.59 µg/mL, respectively, using selenium nanoparticles synthesized with *Crocus capsius* extract [36]. SeNPs fabricated using other plant extracts, such as *Withania somnifera* [29], *M*. *oleifera* [37], *Cassia auriculata* [47], *Hibiscus esculentus* [98], or *Mentha longifolia* [110], also demonstrated high antitumor abilities, inducing apoptosis of a variety of cancer cells with the promotion of reactive oxygen species. In addition, the combination of SeNPs with chemotherapy drugs used to treat cancer can enhance their cytotoxic effects. Additionally, the combination of SeNPs with some chemotherapy drugs used to treat cancer enhanced their cytotoxic effects [109,110]. The study by Dana et al. [111] displayed that chitosan-coated SeNPs could enhance the sensitivity of 5-fluorouracil against glioma, one of the most aggressive cancers, probably by promoting the internalization of nanoparticles via endocytosis. Thus, taking into account the antioxidant properties and cytotoxic effects of plant-based selenium nanoparticles, they can be designed and used as effective chemotherapeutic drugs with the desired size, loading capacity, and controlled release [112,113,114].

The antioxidant activities of SeNPs synthesized with mandarin peel-derived pectins were screened using two chemical-based screening tests, the Trolox radical scavenging activity (TEAC) method and the Folin–Ciocalteu (FC) assay, which determine the reducing properties of a sample [22]. The results from both tests indicated that functionalization with polyphenols derived from olive pomace significantly increased the antioxidant properties of SeNPs (Figure 4A,B). To provide better insight into the biological relevance of SeNP antioxidant activity, cell-based methods were also applied with HepG2 and Caco-2 cell lines in the MTT test for cell viability following treatment with tBOOH as a pro-oxidant (Figure 4C,D). The most visible positive effect was observed for SeNPs coated with pure pectin in the Caco-2 test and for pure pectin with the addition of olive pomace (HepG2 test).

### 3.2. Antimicrobial Activity

The antimicrobial activity of SeNPs (which can be antibacterial, fungicidal, or antiviral) is based on different mechanisms, both intracellular and extracellular, such as the formation of reactive oxygen species, penetration to the bacterial membrane and disruption of phospholipids in the cell wall, and inactivation of proteins that result in bacterial lysis [11,13,19,115]. These activities are size-dependent since smaller nanoparticles can easily cross the cell wall and membrane.

Several papers presented the antibacterial activities of plant-based selenium nanoparticles against both Gram-positive (*Staphylococcus aureus*, *Streptococcus mutants*, *Bacillus subtilis*, *Corynebacterium diphtheriae*) and Gram-negative (*Proteus* sp., *Escherichia coli*, *Klebsiella pneumonia*, *Pseudomonas aeruginosa*, *Salmonella typhimusium*) bacterial strains (Table 2). SeNPs are more effective against Gram-negative bacteria because of their thin peptidoglycan cell wall [23,33,44,59]. SeNPs also exhibit activity against fungal plant pathogens (*Fusarium oxysporum*, *Colletotrichum gloesporioides*), which are typically associated with hospital-acquired infections and food spoilage, posing a major threat to human health [116,117]. SeNPs using leaf extract of *Withania somnifera* exhibited significant activity against *K. pneumonia* and *B. subtilis*, but no activity against *E. coli* strains [31]. It should be noted that SeNPs synthesized by a chemical method using ascorbic acid as a reductor also did not show an antibacterial effect against this anaerobic Gram-negative bacterium, unlike many other nanoparticles produced with herbal polyphenols [27]. The presence of several phytochemicals on the nanoparticle surface may enhance the antimicrobial potential of plant-based SeNPs [13,19]. They can inhibit bacterial biofilm formation, even dual-species biofilms that protect antibiotic treatments [117,118].

Serov et al. [118] noted that the value of the minimum inhibitory concentration (MIC) depends on the method of SeNP synthesis. Using physical synthesis methods, such as laser ablation or microwave irradiation, MIC values for effective antibacterial action did not exceed 100 μg/mL. On the other hand, when using microwave generation of nanoparticles, the MIC values were approximately 100–300 μg/mL, which was significantly worse than nanoparticles obtained by other methods.

Figure 5 shows that a higher zone inhibition for *E. coli* was observed using SeNPs synthesized with *Vaccinium arctostaphylos* fruit extract, compared to ciprofloxacin (used to treat many bacterial infections but with some side effects) [49]. Biogenic SeNPs from lemon juice possessed a higher bactericidal effect against several strains compared to the nanoparticles obtained using grapefruit juice [50].

Plant-based SeNPs also have antifungal activities at low concentrations, reducing possible adverse effects on human health [27,119,120,121]. These properties of SeNPs synthesized using the extract of *Amphipterygium glaucum* leaves against plant pathogen *Fusarium oxysporum* were observed at SeNP concentrations of 0.25–1.7 mg/mL [27], while AgNPs for reducing the abundance of these fungi were efficient under at concentration of 150 mg/L, with the minimum inhibitory concentration of 75 mg/L [122]. The antibacterial and antifungal activity of SeNPs can be used for coating surface medical devices to prevent biofilm formation and in several wrapping paper products used in the food industry [123,124].

### 3.3. Antidiabetic Activity

Diabetes mellitus is characterized by high levels of glucose in the blood (hyperglycemia), deficiency of insulin secretion, or its resistance, leading to several complications and serious health problems. The antidiabetic properties of SeNPs have been mostly associated with oxidative stress, inflammation, hyperlipidemia, and also with dysregulated metabolic syndrome [125,126]. SeNPs synthesized using plant extracts due to their high antioxidant activity and bioavailability, as well as the low risk of their excess, have been evaluated for the prevention and treatment of diabetes [127,128,129,130,131]. It was reported that Se-NPs synthesized with some flavones (luteolin and its glycoside diosmin) had good potential to reduce the disorders of diabetes mellitus [127]. These SeNPs (average diameter of 47.84 nm, zeta potential value of −17.6 mV) regulated blood glucose, glycogen, glycosylated hemoglobin, the lipid profile, and increased insulin production from pancreatic cells. Also, SeNPs functionalized with naringenin, a flavonoid belonging to the flavanone subclass, and baicalin (glycosyloxyflavone) exhibited antidiabetic activities [128]. Nanoparticles with a size diameter of 80–119 nm and zeta potential of −22.3 mV formed a stable dispersion with stability for five months. The treatment of streptozotocin-induced mice with selenium nanoparticles covered with those compounds improved the regulation of hepatic glucose, increased insulin levels, and reduced alterations in lipoproteins and lipids. Functionalized SeNPs had greater antidiabetic activity than uncoated Se nanoparticles and extract alone. It suggested that this effect can be due to the synergic action of flavonoids and selenium [128].

## 4. Conclusions

Green methods for synthesizing SeNPs are considered to be single-step, inexpensive, and eco-friendly processes. The primary objective of selenium nanoparticle synthesis is the creation of particles with the smallest possible size and the highest possible stability [132]. The use of plant extracts is more favorable compared to the bacterial path as it eliminates tedious procedures and the cost of maintaining cell cultures. In addition, several waste materials from the agricultural industry can be utilized. One of the advantages of plant-mediated synthesis is the stability achieved due to natural stabilizers, which can be either the metabolites or biomolecules of that particular source. However, the capped nature of SeNPs makes them less useful in electrical and catalysis applications [133]. Thus, researchers who use plant extracts for their synthesis tend to focus on biomedical applications.

The size, dispersion, content, and surface properties of SeNPs are essential factors when they are used in diverse fields of biotechnology. Plant-mediated synthesis methods produce a wide range of particle sizes (Table 1). According to Serov et al. [119], only physical methods for SeNP synthesis, such as laser ablation or microwave, allow for the production of nanoparticles with narrow size distribution. Researchers have shown that an increase in the concentration of the extract increases the selenium nanoparticle size while decreasing the polydispersity index. When a selenium precursor is present in smaller amounts, nanoparticles grow in size. Surface capping and functionalization increase the diameter of nanoparticles but offer steric stabilization. It should be noted that the mixture of different compounds present in plant extracts could affect the mechanism of a synthesis reaction and, consequently, the properties of nanoparticles, such as homogeneity.

SeNPs have shown higher biocompatibility and lower toxicity than other inorganic or organic Se species. Moreover, they can be functionalized and stabilized with different compounds or loaded with specific drugs. Many studies revealed their strong antioxidant, anticancer, antidiabetic, and antimicrobial activities. SeNPs have found several biomedical applications in diagnosis, treatment, monitoring, and drug delivery systems. Their antioxidant and antibacterial properties can be utilized in the food and pharmaceutical industry. SeNPs may have also potential commercial applications, such as in nutritional supplements for humans and veterinary needs [102,134], the development of a test system for the combined detection of anti-SARS-CoV-2 IgM and IgG in human serum and blood [135], as a component of an alternative biodegradable biopolymer that can be used as active food packaging material [136], or in the production of nanoparticle fertilizers for crop production [137].

The biological properties of SeNPs have only been evaluated in model or animal studies, limiting the research scale. Published data report that selenium supplementation in the nanoform lowers the risk of its excess and no significant toxic effects of SeNPs were found in rats [138], zebrafish [139], or the shrimp hemocyte [140]. More clinical trials are required to gain more information and to expand SeNP applications in human health. However, this effect depends on the dose manner and time of exposure [13]. Probably also on whether SeNPs are to be used to prevent disease or as a therapeutic agent. For example, the relationship between inorganic and organic selenium species and diabetes mellitus is best represented in a dose-dependent manner, yielding a U-shaped graph [5,8]. Determining a similar relationship for the toxicity of SeNPs would be very helpful in pharmaceutical applications and nutritional supplements.

## Figures and Tables

**Figure 1 materials-17-03748-f001:**
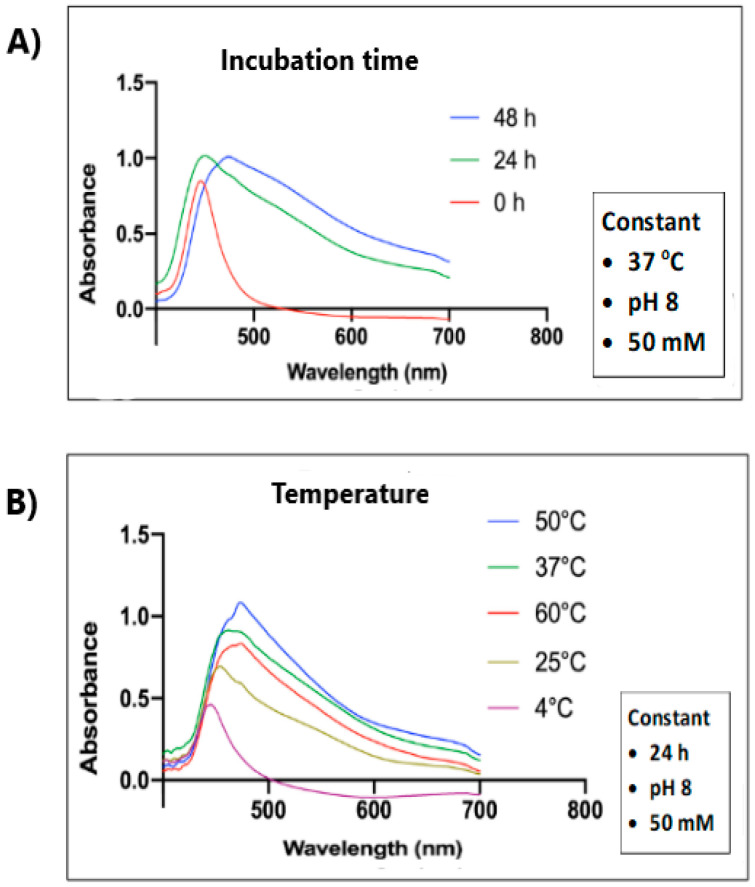
UV–Vis spectra of SeNPs synthesized using *Moringa oleifera* extract as a function of (**A**) incubation time and (**B**) temperature. Reproduced under terms of the CC-BY license [37]. Copyright 2023, MDPI.

**Figure 2 materials-17-03748-f002:**
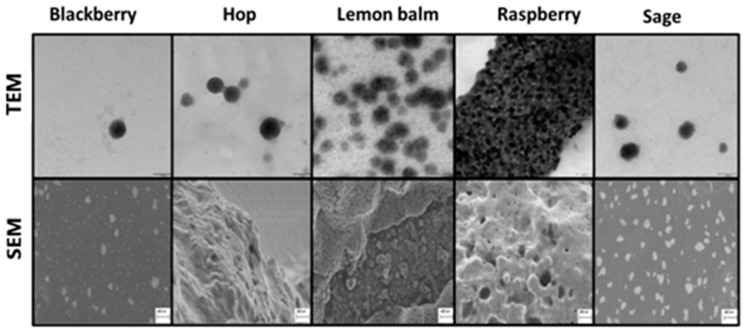
TEM and SEM images of SeNPs obtained using herb extracts. Reproduced under terms of the CC-BY license [27]. Copyright 2024, MDPI. Scale bar: 1 cm = 1000 nm.

**Figure 3 materials-17-03748-f003:**
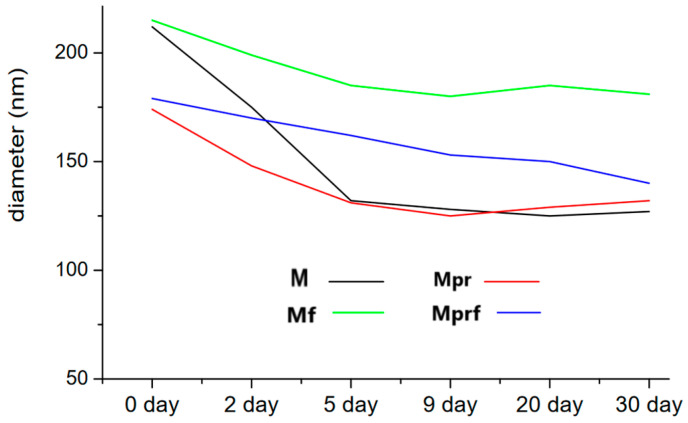
Stability of SeNPs modified by polyphenols and pectins during 30 days of storage regarding average diameter. Reproduced under terms of the CC-BY license [23]. Copyright 2023, MDPI.

**Figure 4 materials-17-03748-f004:**
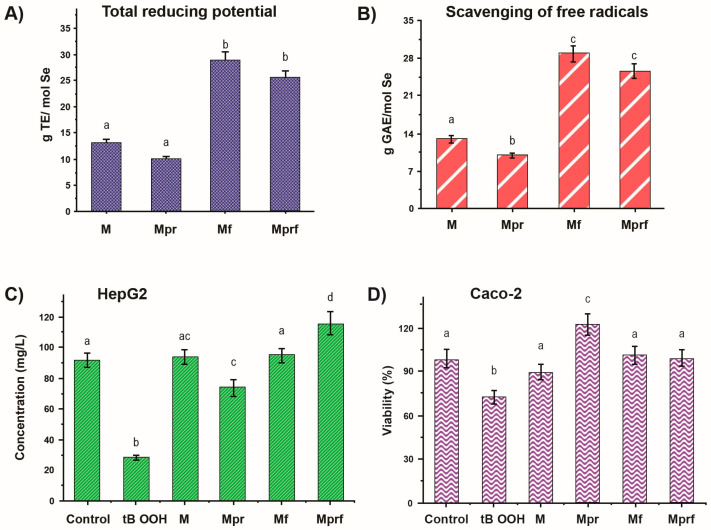
Stability of SeNPs stabilized with raw and purified mandarin pectins during 30 days of storage regarding: (**A**) total reducing potential using FC assay; (**B**) radical scavenging activity with TEAC method; antioxidant activity in HepG2 (**C**) and Caco-2 (**D**) cell lines in the MTT test. Abbreviations: **M:** SeNPs stabilized with raw pectin; **Mpr:** SeNPs stabilized with purified pectin; **Mf:** SeNPs stabilized with raw pectin and functionalized with olive pomace extract; **Mprf:** SeNPs stabilized with raw pectin and functionalized with olive pomace extract. The data are presented as the mean ± standard deviation of four parallel investigations. Data marked with different letters indicate significant differences (*p* ≤ 0.05). Reproduced under terms of the CC-BY license [23]. Copyright 2023, MDPI.

**Figure 5 materials-17-03748-f005:**
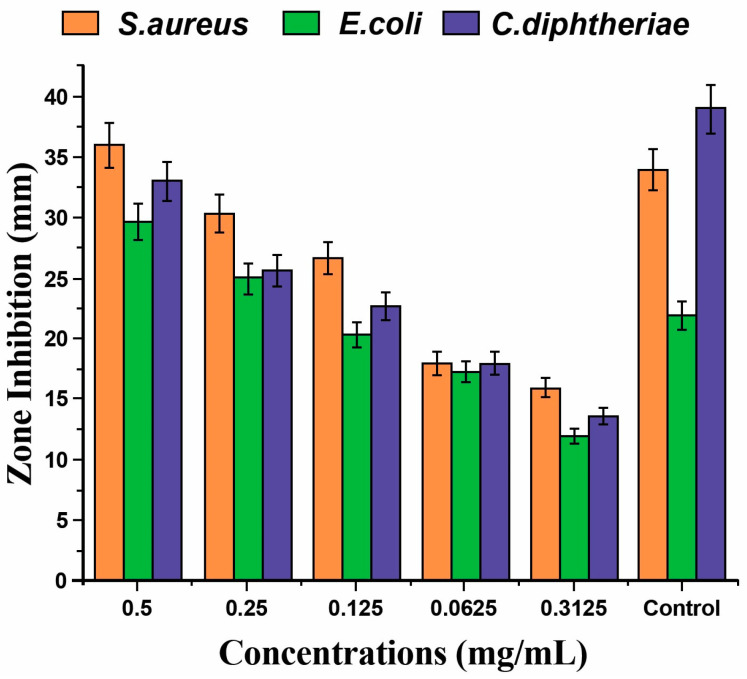
The zone inhibition effect of different concentrations of SeNPs synthesized with *Vaccinium arctostaphylos* fruit extract against *S. aureus*, *E. coli*, and *C. diphtheriae* strains [41]. Ciprofloxacin was used as a control. Reproduced under terms of the CC-BY license [49]. Copyright 2023, Taylor & Francis.

**Table 2 materials-17-03748-t002:** Antimicrobial activity of SeNPs using plant-mediated synthesis.

Strain	Plant Material	Activity	Ref.
Gram-positive bacteria			
*Staphylococcus aureus*	*Abelmoschus esculentus*	MIC = 32 µg/mL	[22]
	*Withanua somnifera*	ZOI = 19.66 mm	[34]
	*Crocus caspius*	MIC = 17.08 µg/m	[36]
	Coffee beans	ZOI = 8 mm	[42]
	*Pluchea indica*	MIC = 31.25 µg/m	[44]
	*Olea ferruginea*	MIC = 11.33 µg/mL	[48]
	*Vaccinium arctostaphylos*	ZOI = 36 mm	[49]
	*Onion*	MIC = 10.67 µg/mL	[53]
	*Dispros montana*	ZOI = 34.16 mm	[59]
*Bacillus subtilis*	*Withanua somnifera*	ZOI = 12 mm	[30]
	*Pluchea indica*	MIC = 3.9 µg/m	[44]
	*Cassica auriculata*	ZOI = 27 mm	[47]
	*Olea ferruginea*	MIC = 11.33 µg/mL	[48]
	Grapefruit juice	ZOI = 19 mm	[50]
	Lemon juice	ZOI = 24 mm	[50]
	*Dispros montana*	ZOI = 44.14	[59]
*Streptococcus mutants*	*Abelmoschus esculentus*	MIC = 128 µg/mL	[22]
*Enterococcus faecalis*	*Crocus caspius*	MIC = 136.66 µg/mL	[36]
*Corynebacterium diphtheriae*	*Vaccinium arctostaphylos*	ZOI = 25.77 mm	[49]
*Acinetobacter baumannii*	*Crocus caspius*	MIC = 17.08 µg/mL	[36]
*Micrococcus luteus*	Grapefruit juice	ZOI = 18 mm	[50]
	Lemon juice	ZOI = 22 mm	[50]
Gram-negative bacteria			
*Escherichia coli*	*Abelmoschus esculentus*	MIC = 256 µg/mL	[22]
	*Crocus caspius*	MIC = 68.33 µg/mL	[36]
	*Coffee beans*	ZOI = 7.1 mm	[42]
	*Pluchea indica*	ZOI = 20.2 mm	[44]
	*Cassica auriculata*	ZOI = 29 mm	[47]
	Grapefruit juice	ZOI = 19 mm	[50]
	Lemon juice	ZOI = 24 mm	[50]
*Klebsiella pneumonia*	*Withanua somnifera*	ZOI = 12.0 mm	[34]
	Grapefruit juice	ZOI = 20 mm	[50]
	Lemon juice	ZOI = 24 mm	[50]
	*Dispros montana*	ZOI = 48.0 mm	[59]
*Pseudomonas aeruginosa*	*Abelmoschus esculentus*	MIC = 128 µg/mL	[23]
	*Crocus caspius*	MIC = 34.17 µg/	[36]
	*Pluchea indica*	MIC = 15.62 µg/m	[44]
*Proteus mirabilis*	*Crocus caspius*	MIC = 136.66 µg/mL	[36]

MIC: minimal inhibitory concentration; ZOI: zone of inhibition.

## Data Availability

No new data were created or analyzed in this study.

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
