# Peer review of "Plant Extracts for Production of Functionalized Selenium Nanoparticles"

_materials, 2024, doi:10.3390/ma17153748_

Round 1

Reviewer 1 Report

Comments and Suggestions for Authors

This manuscript presents a comprehensive review on green methods for SeNPs synthesis using plant extracts. It discusses the effect of synthesis conditions on the characteristics of the obtained SeNPs. Albeit the topic is in general of great interest, the novelty is limited, and the following issues should be considered before consideration for publication:

1.       The quality of figures 1 and 4 should be improved. Numbers are unreadable.

2.       The introduction should include the electrodeposition as a versatile method to synthesize Se nanoparticles and thin film by referring to 10.1016/j.jelechem.2019.01.015 and 10.1007/s10008-024-05807-8

3.       In table 1, Na2SeO3 was almost used in all cases. The mechanism of its reduction to Se(0) using plant extract should be added.

4.       Relevant works on the use of NPs in emerging applications and their advantages can be considered in the first paragraph of the introduction such as these ones (Facet-Dependent Intrinsic Activity of Single Co3O4 Nanoparticles for Oxygen Evolution…) and (Electrodeposited cobalt nanosheets on smooth silver as a bifunctional catalyst for OER and ORR…)

5.       The introduction and discussion should be strengthened by adding a short text on the crucial role of the extract in stabilizing and capping NPs and the importance of identification of the components of the plant extract to correlate them to the properties and reactivities of the synthesized particles e.g. this work (Eco-friendly polymer succinate capping on silver nano-particles for enhanced stability…) (Tannic acid capped gold nanoparticles: capping agent chemistry controls the redox..) and these references 10.1149/1945-7111/acfa69, 10.3390/molecules29040801

6.       The discussion on the application of SeNPs should be better organized into subtitles and extended by giving an example of each application. There has been a recent review on this topic “A review on selenium nanoparticles and their biomedical applications”. Author should be careful to avoid overlap.

7.       A table comparing the antibacterial activity and the antioxidant activity of different Se particles would be interesting or adding an extra column with this information in Table 1.

8.       A brief summary on the mechanism of antibacterial and antioxidant activity of SeNPs should be added.

9.       The language should be revised. Some typo errors and language flaws are present e.g. L504 (.. is the best represented..)

Comments on the Quality of English Language

minor editing

Reviewer 2 Report

Comments and Suggestions for Authors

The review of SeNPs is comprehensive and interesting for the broad readership. Only one comment is about the figures cited in this review paper. The resolution of these figures should be extensively improved.

Reviewer 3 Report

Comments and Suggestions for Authors

The paper

 Plant Extracts for Production of Functionalized Selenium Nanoparticles

conducted by

 Krystyna Pyrzynska provides information about SeNPs synthesized by Green methods.

The present investigation offers a valuable selection of information regarding the preparation conditions of SeNPs, as well as their influence on the critical characteristics (particle size, polydispersity index, surface charge, etc.).

The subject is relevant for research, because it brings valuable information about SeNPs synthesized by Green methods that are considered inexpensive, and eco-friendly.

A synthesis of SeNPs applications is presented, at the same time, a comparative analysis of SeNPs prepared by green synthesis with different plant extracts, regarding both the characteristics and the effects demonstrated in vitro / in vivo.

The references are recent and relevant to the information presented.

However, I request the author to make some clarifications concerning figures, as follows:

I understand that figure 3 shows images of SeNPs published by other authors, and figures 2, 4 and 5 show certain characteristics of them. If the figures are taken from other works, in order to present the information and the results of previous research as clearly as possible, I ask the author to give up the figures and extract the important data, synthesizing them in tables.

Reviewer 4 Report

Comments and Suggestions for Authors

Dear Editor of Journal of Materials,

Thanks to the author to provide an interesting review entitled Plant Extracts for Production of Functionalized Selenium Nanoparticles. The study aligns with the scope of the Journal of Material, making it potentially interesting to its readers. References are updated and cover the scope of review focus, but the authors left a lot of sentences and claim without citations and in other place used the bunch of citations, (please see Major issue). Moreover, mistakes appear in citations order, for instance Golub et al, are listed as 48 in bibliography but in text appeared 48 and 49. All citation should be re-check with high attentions. Therefore, this paper in this stage have a fundamental concern which should be address and I try to explain and guide in details. It would be advisable to undergo a Major Revision, Resubmission. I keep the right to recommend rejection to the journal editor by including comments from other reviewers.

Major Issues

1.       Based on the title, there is a noticeable absence of synthesis procedures for producing functionalized selenium nanoparticles using plant extracts. At this stage, the content of the text does not adequately address the title.

2.       Although the author used updated and relevant citations, their positions and dispersion within the text do not make sense. The authors leave many statements without citations and, in other instances, use a cluster of citations, which is not acceptable. Any criteria and claim made in the text should be supported by related citations, and avoid the bunch of citations to cover general aspects. For instance, lines 124-126, needs to be revised. Each citation should be explained separately and clearly. I highly recommend addressing this issue consistently throughout the entire manuscript.

3.       Some subheadings do not cover the content of the corresponded text. The author explains certain criteria and provides examples, but this left the text without any clear correlation between the issues mentioned. For instance, lines 283-285 ('What was the zeta?! How is it related to stability?! What does this example aim to show?!'), lines 286-288 ('What is the correlation between pH and zeta?! How does it work?'), and why line 291 talk about stabilizing agents in a section related to zeta? How is it correlated with zeta? What does the author intend to explain and how is it related to zeta?"

4.       It appears that the author used figures from other publications without citing them or declaring the copyright terms and conditions (this should be verified by the editorial team). However, the majority of figures have extremely low quality and need to be replaced.

5.       I would suggest adding more physicochemical properties related to selenium nanoparticles, before discussing applications. For instance, the effect of plant extracts on the thermal and chemical stability of selenium nanoparticles, as well as their solubility and dispersibility.

6.       I would suggest dividing the applications into subheadings such as antioxidant activities, antimicrobial effects, cancer therapy, etc.

Minor Issue

1.       Line 26, please remove extra and before optical.

2.       Line 55, please add citations after `and biodegradable.´

3.       Line 63, please add citation after `to the corresponding quinones.´

4.       Line 65, please add citations after `and many other properties.´

5.       Line 67, please add citations after `and antiparasitic properties

6.       Line 70, please add citations after `group of amino residues

7.       Figure 1:

                                i.            please draw the structure in chemdraw, it is not allowed to use the figures from web or …

                              ii.            The quality is too low, which we be solved by drawing the structures;

                            iii.            It is highly recommended to add the example of all structures mentioned in previous paragraph, where figure 1 was cited, for example one example from protein;

                            iv.             Please divide the names.

8.       Line 99, please add citations at the end of paragraph.

9.       Line 126, I would recommend expanding the list of papers cited here. Citations without expanding on the prospects and ideas do not make sense in any type of paper, particularly review papers. Therefore, I suggest expanding all citations in this paragraph.

10.   line 137, please add citations after `more specific properties´.

11.   line 152-153, please use the similar format used for other (for example: and ginger at 550 nm).

12.   Line 144, The subheading 'Particle Size and Distribution' should focus on size and poly dispersity which is distribution. I recommend separating the UV-Vis content from the size and distribution section into a distinct section with a related subheading, since there are related placed before Section 2.1.

13.   line 185, By adding the SEM and TEM content in Section 2.1, please change the subheading to 'Section 2.1. Particle Size, Distribution, and Morphology'. Moreover, the distribution criteria are related to the PDI value, which is not mentioned in the content of Section 2.1. Please revise the subheading title and remove 'distribution.' Consider renaming it to 'Section 2.1. Particle Size and Morphology.' Alternatively, you can combine all related content under one subheading that includes all parameters. For instance, 'Section 2.1. Particle Size, distribution, and Morphology.' subsequently removed subheading 2.2. Poly dispersity index.

14.   Line 240, please remove extra `the´.

15.   Line 248, missing citations for all claimed criteria.

16.   Line 268-275, are not related to Section 2.3. Moreover, line 271 mentions a critical claim that is left without citations. If the presence of certain elements affects the surface charge, the author should expand on these criteria and explain which elements cause what charges, how these charges are calculated or measured, and the correlation with X-ray analysis. The author should end with some examples that specify the use of FTIR and X-ray to estimate or analyze the surface charge of nanoparticles.

17.   line 277-271, in my opinion, this should be placed at the beginning of this section.

18.   line 283-285, it presents a good example, but what is its relations with zeta, charge?

19.   line 296, The explanation of chitosan is too extensive for a section discussing surface charge. A brief explanation in parentheses with a single citation is sufficient, or this part could even be removed. I believe this section should focus on zeta potential and its effects on delivery, absorption, etc.

20.   Line 305-307, what is the correlation with zeta and charge? I highly recommend rewriting this section with careful attention. Try to clearly connect the section's aim to what you have claimed and cited here.

21.   Please avoid using a bunch of citations at lines 363, 371, 395, 428, etc. Instead, try to expand and make connections between the application and its use, based on the advantages of plant extracts in the production of selenium nanoparticles.  

I would highly recommend to take care of all issues explained as minor and major issue with high attentions throughout entire text.

Good luck.

Comments on the Quality of English Language

English was fine. 

Round 2

Reviewer 1 Report

Comments and Suggestions for Authors

Author has addressed some of the comments, but most of the points weren't considered as follow:

1. The mechanism of reduction of Na2SeO3 to Se(0) using plant extract isn't mentioned in L53-64. Equations or schematic should be included.

2. My comment number 2 in the previous report about electrodeposition method wasn't supported by suggested references. L37-39 have no references.

3. Although the review is focused on SeNPs, one sentence on the general applications of NPs can be added. You have already cited papers not on SeNPs in the first paragraph, and this can be extended.

4. My previous comment number 5 wasn't properly considered and L67 isn't supported by the suggested references.

Comments on the Quality of English Language

is fine

Reviewer 3 Report

Comments and Suggestions for Authors

The work has been significantly improved. Important data regarding the characteristics and applications of NPs have been synthesized in tables that allow a clear understanding of the informations. Also, they were discussed extensively in the text.

Author Response

Thank you for your opinion and comments.